# The effect of excessive trabeculation on cardiac rotation—A multimodal imaging study

**Kinga Grebur**[1], **Balázs Mester**[1], **Márton Horváth**[1], **Kristóf Farkas-Sütő**[1], **Zsófia Gregor**[1], **Anna Réka Kiss**[1], **Attila Tóth**[1], **Attila Kovács**[1], **Alexandra Fábián**[1], **Bálint Károly Lakatos**[1], **Bálint András Fekete**[1,2], **Katalin Csonka**[2], **Csaba Bödör**[2], **Béla Merkely**[1], **Hajnalka Vágó**[1], **Andrea Szűcs**[1]*

1 Heart and Vascular Center of Semmelweis University, Budapest, Hungary, 2 Department of Pathology and Experimental Cancer Research, Semmelweis University, Budapest, Hungary

* szucsand@gmail.com

**Data Availability Statement:** All relevant data used for statistical analyses are available as Supporting Information file 3 of the manuscript.

## Abstract

### Background

Cardiac rotational parameters in primary symptomatic left ventricular noncompaction (LVNC) with preserved left ventricular ejection fraction (LVEF) are not well understood. We aimed to analyze cardiac rotation measured with cardiac magnetic resonance feature-tracking (CMR-FT) and speckle-tracking echocardiography (Echo-ST) in LVNC morphology subjects with preserved LVEF and different genotypes and healthy controls.

### Methods

Our retrospective study included 54 LVNC subjects with preserved LVEF and 54 control individuals. We evaluated functional and rotational parameters with CMR in the total study population and with echocardiography in 39 LVNC and 40 C individuals. All LVNC subjects were genotyped with a 174-gene next-generation sequencing panel and grouped into the subgroups: benign (B), variant of uncertain significance (VUS), and pathogenic (P).

### Results

In comparison with controls, LVNC subjects had reduced apical rotational degree (p = 0.004) and one-third had negative apical rotation. While the degree of apical rotation was comparable between the three genetic subgroups, they differed significantly in the direction of apical rotation (p<0.001). In contrast to control and B groups, all four studied cardiac rotational patterns were identified in the P and VUS subgroups, namely normal rotation, positive and negative rigid body rotation, and reverse rotation. When the CMR-FT and Echo-ST methods were compared, the direction and pattern of cardiac rotation had moderate to good association (p<0.001) whereas the rotational degrees showed no reasonable correlation or agreement.

**Funding:** The research was supported by the ÚNKP-23-3-II new national excellence program of the Ministry for Culture and Innovation (https://nkfih.gov.hu/for-the-applicants) from the source of the National Research, Development and Innovation Fund; by the Development of Scientific Workshops of Medical, Health Sciences and Pharmaceutical Education (Project identification number: EFOP-3.6.3-VEKOP-16-2017- 00009); by the TKP2021-NKTA-46 implemented with the support provided by the Ministry of Innovation and Technology of Hungary (https://2015-2019.kormany.hu/en/ministry-for-innovation-and-technology) from the National Research, Development and Innovation Fund, financed under the a TKP2021-NKTA funding scheme; and by the European Union project RRF-2.3.1-21-2022-00004 within the framework of the Artificial Intelligence National Laboratory, Hungary (https://mi.nemzetilabor.hu/). KG received funding from the ELTE Márton Áron Research Fellowship program (https://www.elte.hu/en/). The sponsors of funders do not play any role in the study design, data collection and analysis, decision to publish, or preparation of the manuscript.

**Competing interests:** The authors have declared that no competing interests exist.

## Conclusion

While measuring cardiac rotation using both CMR-FT and Echo-ST methods, subclinical mechanical differences were identified in subjects with LVNC phenotype and preserved LVEF, especially in cases with genetic involvement.

## 1. Introduction

In healthy, mature myocardium, clockwise (CW) basal and counterclockwise (CCW) apical end-systolic endocardial rotation characterizes the normal rotational pattern, resulting in a suction effect and contributing to the left ventricular function [1, 2]. Several studies evaluated cardiac rotation using different imaging techniques, such as speckle-tracking echocardiography (Echo-ST) and cardiac magnetic resonance imaging feature-tracking (CMR-FT); both validated against the gold standard tagging magnetic resonance imaging [3–6].

Other cardiac rotational patterns have also been described, e.g. reverse rotation, which is the opposite of normal rotation; and rigid body rotation (RBR), where the base and apex rotate in the same negative (CW) or positive (CCW) direction (**Fig 1**) [1]. Although these patterns may assume abnormal cardiac maturation [7], RBR has been described in the literature in various cardiovascular diseases and also in healthy individuals [2, 8–11]. Moreover, it has been most strongly associated with left ventricular noncompaction (LVNC) morphology [12–15].

Among the above-mentioned rotational patterns, particularly the negative RBR was associated with impaired left ventricular (LV) ejection fraction (EF) and cardiovascular complications in excessive trabeculation [10, 11, 13, 16]. Recognizing the wide clinical spectrum and genetic heterogeneity of LV hypertrabeculation ranging from healthy individuals to end-stage heart failure [17, 18], analysis of cardiac rotation could contribute to a better understanding of LVNC morphology. Therefore it could be of particular interest, especially since cardiac rotation was not investigated before in primary LVNC with various genetic background and preserved LVEF.

In this study, we aimed to characterize the cardiac rotation of symptomatic LVNC morphology subjects with preserved LVEF and different genotype and control (C) individuals; and to evaluate the association of cardiac rotation with functional values and LVNC genotype. Our

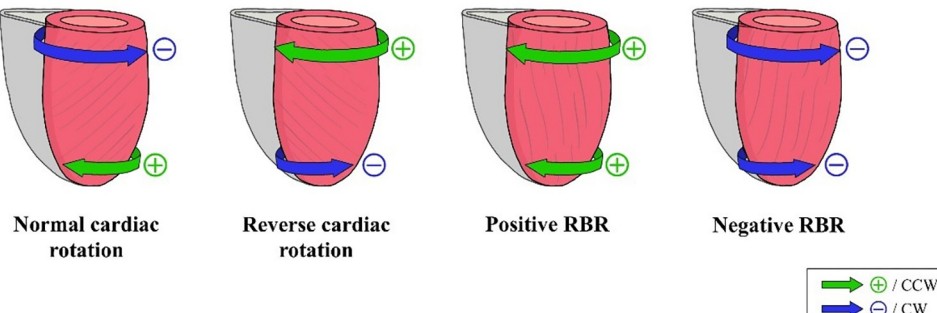

**Fig 1. Cardiac rotational patterns.** The normal cardiac rotational pattern is characterized by negative or CW basal and positive or CCW apical peak end-systolic rotation, and the reverse cardiac rotation is represented by positive or CCW basal and negative or CW apical rotation. RBR is defined by rotating the basal and apical parts in the same direction in a single patient: positive RBR when this rotation is positive, and negative RBR when it is negative. CCW = counterclockwise, CW = clockwise, RBR = rigid body rotation.

further aim was also to assess the intermodality agreement of the rotational parameters measured using the CMR-FT and Echo-ST methods.

## 2. Material and methods

### 2.1 Study population

Fifty-four symptomatic LVNC subjects with preserved LVEF (CMR LVEF: 64.7±6.0%) and 54 sex- and age-matched healthy C individuals (CMR LVEF: 69.3±4.8%) from a Caucasian population were included in our retrospective study. All participants underwent CMR examination; the baseline characteristics and clinical data are listed in **Table 1a**.

Subjects with isolated, persistent hypertrabeculated phenotype who fulfilled both the Petersen (noncompact/compact myocardial layer ≥ 2.3) and Jacquier LVNC criteria (noncompact myocardial mass > 20% of the total myocardial mass), had an LVEF ≥ 50% measured on CMR images, and available cardiogenetic data were included in the **LVNC group** [19, 20]. We excluded from this study LVNC patients with reduced LVEF (< 50%) on CMR image and subjects with transient or secondary hypertrabeculation, congenital, ischemic or valvular heart diseases, other or overlapping cardiomyopathies (CMP), relevant comorbidities e.g. diabetes mellitus, hypertension or chronic kidney disease.

The **C group** included healthy volunteers without cardiovascular or extracardiac disease, based on a detailed medical history and ECG.

Physical activity exceeding 6 hours per week and technical reasons e.g. artifacts, implanted cardiac devices or CMR contrast agent administration prior to segmentation were also exclusion criteria in both study groups [21, 22].

All the procedures performed in this study were in accordance with the 1964 Helsinki Declaration and its subsequent amendments or comparable ethical standards. Ethical approval was obtained from the Central Ethics Committee of Hungary, and all participants provided written informed consent. Data were fully anonymized, and only the first author had access to information that could identify individual participants during and after data collection.

### 2.2 Genetic testing

The genetic testing of LVNC subjects was performed previously with a 174-gene next-generation sequencing panel (TruSight Cardio Sequencing Kit, Illumina, CA, USA) containing genes associated with cardiac diseases (covering 571,897 nucleotides and 3,251 exons). The variant categorisation and clinical relevance analysis was updated by a cardiogenetics specialist (BAF) when patients were enrolled in the study using the online genetic databases e.g. Franklin, NCBI—ClinVar, VarSome, ClinGen and OMIM, based on the recommendations of the American College of Medical Genetics and Genomics (ACMG) [23].

According to these databases, LVNC subjects were divided into three genetic subgroups: 15 patients in the pathogenic (P) group with pathogenic or likely pathogenic mutations in CMP-related genes, 27 patients in the variant of uncertain significance (VUS) group with VUS CMP-related mutations and 12 patients in the benign (B) group without mutations in CMP-related genes. Genetic data were accessed for research purposes on 02/09/2023 and are presented in **S1 Table**.

### 2.3 Image acquisition and analysis

CMR scans of the total population were performed using 1.5 T magnetic resonance imaging scanners (Magnetom Aera, Siemens Healthineers, Erlangen, Germany and Achieva, Philips Medical System, Eindhoven, the Netherlands). Retrospectively gated, balanced steady-state

**Table 1. Baseline and clinical characteristics (a), CMR and echocardiography volumetric, functional and muscle mass parameters (b) and CMR-FT and Echo-ST rotational values (c) of the study populations.**

a)

| | Control | LVNC | p | Pathogenic | VUS | Benign | p |
|---|---|---|---|---|---|---|---|
| **CMR study population (n/male)** | 54/33 | 54/33 | 1.000 | 15/7 | 27/18 | 12/8 | 0.402 |
| **Available Echo data (n/male)** | 40/25 | 39/24 | 1.000 | 9/2 | 21/16 | 9/6 | 0.523 |
| **Age (years)** | 38.7± 15.1 | 40.0± 13.9 | 0.834 | 38.5±17.0 | 40.0±12.4 | 42.0±14.1 | 0.813 |
| **BSA (m$^2$)** | 1.9± 0.2 | 2.0± 0.2 | 0.062 | 1.8±0.2 | 2.0±0.2 | 2.0±0.2 | 0.093 |
| **BMI (kg/m$^2$)** | 23.9± 3.4 | 25.3± 4.2 | 0.072 | 23.4±3.3 | 26.3±4.5 | 25.4±4.2 | 0.096 |
| **Positive family history (n)** | NA | 24 | NA | 10 | 13 | 1 | 0.008* |
| **Atypical chest pain (n)** | NA | 20 | NA | 6 | 6 | 8 | 0.030* |
| **Palpitation (n)** | NA | 28 | NA | 8 | 14 | 6 | 1.000 |
| **Syncope (n)** | NA | 5 | NA | 2 | 2 | 1 | 0.834 |
| **Arrhythmia (n)** | NA | 33 | NA | 8 | 18 | 7 | 0.632 |

b)

| | | Control | LVNC | p | Pathogenic | VUS | Benign | p |
|---|---|---|---|---|---|---|---|---|
| **CMR** | **LVEDVi (ml/m$^2$)** | 67.7±10.7 | 74.4±14.6 | 0.007* | 75.5±18.5 | 74.9±13.3 | 72.0±13.0 | 0.807 |
| | **LVESVi (ml/m$^2$)** | 20.8± 4.9 | 26.6± 8.3 | <0.001* | 27.2±10.6 | 26.8±7.4 | 25.2±7.4 | 0.81 |
| | **LVSVi (ml/m$^2$)** | 46.8±7.7 | 47.9±8.9 | 0.508 | 48.2±10.3 | 48.1±8.9 | 46.8±7.5 | 0.893 |
| | **LVEF (%)** | 69.3±4.8 | 64.7±6.0 | <0.001* | 64.6±6.8 | 64.4±6.1 | 65.4±5.4 | 0.902 |
| | **LVTMi (g/m$^2$)** | 65.9±10.6 | 74.4±16.2 | 0.004* | 73.6±18.4 | 77.0±16.6 | 69.7±11.9 | 0.417 |
| | **LVTPMi (g/m$^2$)** | 19.8±4.7 | 25.1±7.1 | <0.001* | 26.4±8.4 | 25.4±7.3 | 22.6±3.9 | 0.369 |
| | **Basal CLT (mm)** | 6.5±0.6 | 6.4±0.7 | 0.515 | 6.1±0.7 | 6.6±0.7 | 6.5±0.8 | 0.081 |
| | **Mid CLT (mm)** | 5.9±0.4 | 5.7±0.7 | 0.070 | 5.3±0.9 | 5.9±0.5 | 5.9±0.5 | 0.009* |
| | **Apical CLT (mm)** | 5.4±0.2 | 4.8±0.8 | <0.001* | 3.9±0.7 | 5,0±0.5 | 5.3±0.3 | <0.001* |
| **ECHO** | **LVEDVi (ml/m$^2$)** | 66.4±13.6 | 73.0±15.1 | 0.047* | 67.5±19.3 | 75.0±15.0 | 73.5±11.0 | 0.509 |
| | **LVESVi (ml/m$^2$)** | 29.0±7.6 | 34.6±7.7 | <0.001* | 31.2±9.0 | 35.6±8.1 | 35.4±5.4 | 0.389 |
| | **LVSVi (ml/m$^2$)** | 38.3±7.3 | 38.4±7.9 | 0.950 | 36.3±10.8 | 39.4±7.4 | 38.1±6.1 | 0.653 |
| | **LVEF (%)** | 57.9±5.0 | 52.7±2.7 | <0.001* | 53.6±3.4 | 52.7±2.6 | 51.8±2.2 | 0.385 |

c)

| | | Control | LVNC | p | Pathogenic | VUS | Benign | p |
|---|---|---|---|---|---|---|---|---|
| **CMR** | **Basal rotation (°)** | -5.1±6.5 | -3.5±7.1 | 0.209 | -1.0±5.8 | -5.0±6.7 | -3.1±8.8 | 0.207 |
| | **Apical rotation (°)** | 12.3±9.3 | 6.7±11.0 | 0.005* | 8.1±10.8 | 3.4±10.1 | 12.2±11.6 | 0.058 |
| | **Net cardiac twist (°)** | 17.9±10.9 | 12.4±9.7 | 0.006* | 10.3±9.8 | 11.8±8.0 | 16.3±12.3 | 0.262 |
| **ECHO** | **Basal rotation (°)** | -5.4±4.2 | -5.0±6.6 | 0.740 | -4.6±6.0 | -5.8±6.4 | -3.5±7.9 | 0.662 |
| | **Apical rotation (°)** | 5.7±5.0 | 3.8±5.5 | 0.106 | 4.9±5.3 | 3.6±6.5 | 3.0±3.0 | 0.769 |
| | **Net cardiac twist (°)** | 11.0±6.2 | 10.0±7.7 | 0.508 | 9.9±7.3 | 10.3±8.9 | 9.1±5.3 | 0.922 |

LVNC = left ventricular noncompaction, VUS = variant of uncertain significance, CMR = cardiac magnetic resonance imaging, Echo = echocardiography, CMR-FT = CMR feature-tracking method, Echo-ST = speckle-tracking echocardiography, n = number of the study group, BSA = body surface area, BMI = body mass index, LVEDVi = left ventricular end-diastolic volume index, LVESVi = left ventricular end-systolic volume index, LVSVi = left ventricular stroke volume index, LVEF = left ventricular ejection fraction, LVTMi = left ventricular total myocardial mass index, LVTPMi = left ventricular trabeculated and papillary mass index, NA = not applicable, CLT = compact layer thickness, * = p<0.05

free precession (bSSFP) cine sequences were performed with short-axis (SA) and two-, three-, and four-chamber long-axis views from base to apex, covering the whole LV and right ventricle. The slice thickness was 8 mm with no interslice gap, and the field of view was 350 mm on average, adapted to body size. The scans were made by the members of our research group, and the fully anonymized CMR data were accessed for research purposes during the study period (between 2020 and 2022).

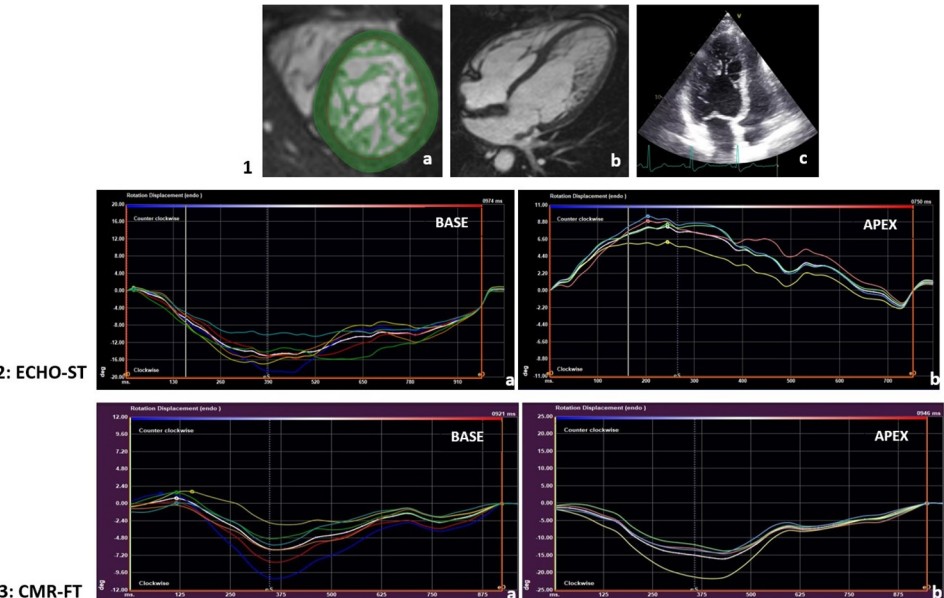

**Fig 2.** Presenting the imaging modalities used for functional analysis: CMR TB technique (1a), four-chamber CMR (1b) and echocardiography images (1c) and the rotational measurements with Echo-ST (2) and CMR-FT (3) methods at basal (2a,3a) and apical (2b,3b) segments. 1. Due to the different signal intensities, the TB algorithm (a) can differentiate the myocardial tissue from the blood volume on CMR SA images and marks it with green color within the green LV epicardial contour (LVTMi). The myocardial tissue (green) recognized within the red LV endocardial contour represents the LVTPMi. Apical LV hypertrabeculation could also be observed on four-chamber view CMR (b) and four-chamber view transthoracic 2D echocardiographic images (c) with prominent trabecular meshwork and deep intertrabecular recesses. 2,3. The interfaces of the CMR-FT and Echo-ST postprocessing software are very similar. The normal rotational pattern, which is presented on a healthy control individual's Echo-ST images (2), is characterized by negative or CW basal (2a) and positive or CCW apical (2b) rotation. Negative RBR is frequently described in hypertrabeculation with negative or CW basal (3a) and apical (3b) rotation, which is illustrated on CMR-FT images of a VUS genotype LVNC morphology subject. CMR = cardiac magnetic resonance imaging, TB = threshold-based method, Echo-ST = speckle-tracking echocardiography, CMR-FT = CMR feature-tracking method, SA = short-axis, LV = left ventricle, LVTMi = left ventricular total myocardial mass index, LVTPMi = left ventricular trabeculated and papillary muscle mass index, CW = clockwise, CCW = counterclockwise, RBR = rigid body rotation, VUS = variant of unknown significance, LVNC = left ventricular noncompaction.

We used Medis Suite software (Medis Suite version 4.1, Medis Medical Imaging Systems, Leiden, the Netherlands) for CMR postprocessing analyses. After automatic tracing and manual correction of the end-diastolic and end-systolic SA cine images from base to apex, we applied the threshold-based (TB) algorithm (MassK module, Medis Suite QMass). The TB method classifies each voxel as blood or myocardium based on different signal intensities: the myocardium on end-diastolic images within the epicardial border forms the total myocardial mass (TM) and within the endocardial border the trabeculated and papillary muscle mass (TPM) (**Fig 2.1a**). The threshold was set to default (50%) and was not modified during the analyses [24]. The LV end-diastolic volume (EDV), end-systolic volume (ESV), stroke volume (SV), and ejection fraction (EF) were also calculated. All parameters were indexed to body surface area (i). As a reference, we considered the normal values by Alfakih and Kawel-Boehm et al [25, 26].

We measured the compact layer thickness (CLT) in two-, three- and four-chamber LA end-diastolic CMR images according to the AHA-17 segment model (excluding the apex-segment 17) and previous studies [19, 27].

For cardiac rotation analyses, we used the feature-tracking algorithm of the commercially available QStrain software (Medis Suite, version 4.1). The CMR-FT algorithm is based on tracking the endocardial border in three directions and following it over time using the LV

endocardial contours on end-diastolic and end-systolic SA images [6, 28]. To analyze the rotational pattern of the LV, according to standard recommendations, the basal slice at the level of the mitral valve and the apical slice well beyond the papillary muscles were selected [1]. For a better standardization of CMR and echocardiographic rotation measurements, we selected the CMR basal and apical slices according to the available echocardiographic images [29].

Thirty-nine out of the 54 LVNC subjects (9, 21 and 9 from P, VUS and B subgroups, respectively) and 40 C individuals also had available and valuable echocardiography images which were performed at the same medical check-up (**Fig 2.1c**). 2D transthoracic echocardiography examinations were performed with a GE Vivid E95 instrument with a 4Vc-D phased-array transducer (GE Vingmed Ultrasound, Horten, Norway). For further analyses, ECG-gated, LV-focused parasternal SA images were obtained at the mitral valve and apical levels, apical long-axis four-, three- and two-chamber view loops with a target frame rate of more than 50 frames per second. After selection of the optimal heart cycle, the 2D TOMTEC Cardiac Performance Analysis software (Philips Ultrasound Workspace, TOMTEC Imaging Systems GmbH, Unterschleissheim, Germany) was used for postprocessing analyses. Manual correction of automatically generated LV endocardial contours was applied on end-diastolic and end-systolic images. Using the long-axis images, we calculated LVEDVi, LVESVi, LVSVi and LVEF functional echocardiographic parameters.

For cardiac rotation analysis, the echocardiography speckle-tracking algorithm was applied to SA images at the mitral valve and apical levels. Echo-ST is based on tracking the intramyocardial speckles between the epi-and endocardial contours in two directions from frame to frame through the cardiac cycle, in contrast to the CMR-FT method, which follows the endocardial border [1, 6, 30].

For the interpretation of CMR-FT and Echo-ST rotational data, the basal and apical endocardial end-systolic peak rotational parameters were evaluated as **quantitative** values in degrees and **qualitative** values as positive (CCW) or negative (CW) direction of rotation [30]. To characterize the overall cardiac rotation, the net cardiac twist parameter was derived as the absolute value of the difference between apical and basal rotation, measured in degrees [1, 29–31]. When analyzing the cardiac rotational pattern, the following conditions were considered: **normal rotation** with CCW apical and CW basal rotation, **reverse rotation** with CW apical and CCW basal rotation, **positive RBR** with CCW apical and basal rotation and **negative RBR** with CW apical and basal rotation (**Figs 1** and **2**).

The interobserver agreement between the two observers regarding functional and rotational CMR parameters (ARK with 6 years and KG with 4 years of experience) and echocardiography data (MH with 5 years and KG with 3 years of experience) was tested on 15 randomly selected study subjects.

## 2.4 Statistical analyses

Continuous parameters are described as mean and standard deviation (SD), and discrete values are described as numbers and percentages. Normal distribution was assessed using the Shapiro–Wilk test and the homogeneity of variances with Levene's test. Differences in continuous parameters between the LVNC and C groups were assessed using independent t-tests if normally distributed and Mann–Whitney U tests if non-normally distributed; and among the genetic subgroups, comparisons were made using one-way analysis of variance (ANOVA) and Tukey's post hoc test in normally distributed variables with equal variances, Welch test and Games-Howell post hoc test in variables with unequal variances, and Kruskal–Wallis test with Bonferroni correction in non-normally distributed data. To compare discrete data, we used the chi-square and Fisher's exact tests. Correlations were assessed with the Pearson correlation

coefficient. Intermodality agreement regarding rotational degrees was tested using the Bland-Altman analysis; Cohen's kappa with chi-square test describes the strength of association with respect to the direction of rotation: <0.2 poor, 0.2–0.4 fair, 0.4–0.6 moderate, 0.6–0.8 good, and >0.8 very good association. The interobserver agreement was tested using the intraclass correlation coefficient (ICC). A p-value <0.05 was considered indicative of statistical significance. Statistical analyses were performed using IBM SPSS Statistics (Version 28.0).

## 3. Results

Interobserver agreement regarding the functional and rotational CMR and echocardiography parameters was moderate to excellent (S2 Table).

Analyzing the CMR and echocardiography functional values, the LVEDVi, LVESVi, LVTMi and LVTPMi were significantly higher; and LVEF and apical CLT were significantly lower in the LVNC population compared to the C group (Table 1b). The CMR-FT apical rotational degree and net cardiac twist were significantly lower in the LVNC than in the C group, while the basal rotational degrees were comparable. Regarding the Echo-ST, we found no significant differences between the two groups in apical and basal rotational degrees and net cardiac twist. Detailed data are shown in Table 1c.

Comparing the direction of rotation, negative (CW) apical rotation was significantly more frequent in the LVNC group than in C individuals with both CMR-FT and Echo-ST methods (CMR-FT LVNC: CCW n = 39, CW n = 15, C: CCW n = 54, CW n = 0, p<0.001; Echo-ST LVNC: CCW n = 29, CW n = 10, C: CCW n = 40, CW n = 0, p<0.001); and it was comparable at the basal level (CMR-FT: LVNC: CCW n = 14, CW n = 40, C: CCW n = 9, CW n = 45, p = 0.240; Echo-ST LVNC: CCW n = 6, CW n = 33, C: CCW n = 2, CW n = 38, p = 0.126).

In the following, we correlated rotational degrees with age and functional parameters in both the LVNC and C groups. In the total LVNC population, apical rotation and net cardiac twist both with CMR-FT and Echo-ST showed a moderate positive correlation with age; and a moderate negative correlation was found between these rotational parameters measured by

**Table 2. Correlation (r) of CMR-FT and Echo-ST rotational parameters with age, functional values and genotype in the LVNC and C groups.**

|  | CMR-FT basal rotation (°) | | CMR-FT apical rotation (°) | | CMR-FT net cardiac twist (°) | | Echo-ST basal rotation (°) | | Echo-ST apical rotation (°) | | Echo-ST net cardiac twist (°) | |
|---|---|---|---|---|---|---|---|---|---|---|---|---|
|  | LVNC | C | LVNC | C | LVNC | C | LVNC | C | LVNC | C | LVNC | C |
| Age (years) | -0.09 | 0.29* | 0.35* | 0.14 | 0.36* | -0.05 | -0.11 | -0.37* | 0.38* | 0.16 | 0.35* | 0.35* |
| LVEDVi (ml/m²) | 0.10 | -0.29* | -0.37* | -0.09 | -0.42* | 0.05 | 0.02 | -0.03 | -0.22 | -0.03 | -0.13 | 0.02 |
| LVESVi (ml/m²) | 0.10 | -0.35* | -0.32* | -0.14 | -0.30* | 0.01 | 0.04 | -0.06 | -0.25 | -0.15 | -0.15 | -0.05 |
| LVSVi (ml/m²) | 0.06 | -0.19 | -0.30* | -0.02 | -0.41* | 0.07 | -0.01 | 0.01 | -0.19 | 0.09 | -0.11 | 0.08 |
| LVEF (%) | -0.07 | 0.24 | 0.13 | 0.13 | 0.04 | 0.01 | -0.17 | 0.07 | 0.09 | 0.31 | 0.14 | 0.17 |
| LVTMi (g/m²) | 0.04 | -0.21 | -0.18 | -0.06 | -0.19 | 0.03 | NA | NA | NA | NA | NA | NA |
| LVTPMi (g/m²) | 0.09 | -0.20 | -0.26 | -0.18 | -0.35* | -0.05 | NA | NA | NA | NA | NA | NA |
| Basal CLT (mm) | -0.02 | 0.04 | 0.11 | -0.04 | 0.05 | -0.07 | NA | NA | NA | NA | NA | NA |
| Mid CLT (mm) | 0.02 | -0.06 | 0.20 | 0.04 | 0.16 | 0.06 | NA | NA | NA | NA | NA | NA |
| Apical CLT (mm) | -0.15 | -0.07 | 0.07 | 0.01 | 0.19 | 0.02 | NA | NA | NA | NA | NA | NA |
| Genotype | 0.12 | NA | -0.11 | NA | -0.21 | NA | -0.06 | NA | 0.12 | NA | 0.04 | NA |

CMR = cardiac magnetic resonance imaging, CMR-FT = CMR feature-tracking method, Echo-ST = speckle-tracking echocardiography, LVNC = left ventricular noncompaction, C = control group, LVEDVi = left ventricular end-diastolic volume index, LVESVi = left ventricular end-systolic volume index, LVSVi = left ventricular stroke volume index, LVEF = left ventricular ejection fraction, LVTMi = left ventricular total myocardial mass index, LVTPMi = left ventricular trabeculated and papillary mass index, r = Pearson correlation coefficient, * = p<0.05, NA = not applicable, CLT = compact layer thickness

CMR-FT and LVEDVi, LVESVi and LVSVi. Moreover, the CMR-FT net cardiac twist also had a moderate negative correlation with LVTPMi. The CMR-FT and Echo-ST basal rotational degree showed no significant correlation. Interestingly, of the abovementioned associations only the positive correlation between Echo-ST apical rotation and age was present in the C group. Data are shown in **Table 2**.

Afterward, we analyzed the connection between cardiac rotation and genotype in our LVNC population. By comparing the P, VUS and B genetic subgroups, functional CMR and echocardiography parameters were comparable, while mid and apical CLT differed significantly (**Table 1b**) and showed moderate to good correlation with genotype (mid CLT r = -0.351, p = 0.009; apical CLT r = -0.691, p<0.001). Regarding the clinical manifestation, positive family history was more frequent in LVNC individuals with genetic involvement, while atypical chest pain in the B subgroup (**Table 1a**).

No significant differences were reported between the three genetic subgroups in rotational degrees measured both with CMR-FT and Echo-ST methods (**Table 1c**); and no significant correlation was found between genotype and quantitative rotational parameters (**Table 2**). Analyzing the direction of rotation, the negative (CW) apical rotation measured with CMR-FT was significantly more frequent in the P and VUS subgroups than in the B subgroup (CMR-FT P: CCW n = 10, CW n = 5, VUS: CCW n = 17, CW n = 10, B CCW n = 12, CW n = 0, p<0.05; Echo-ST P: CCW n = 8, CW n = 1, VUS: CCW n = 13, CW n = 8, B CCW n = 8, CW n = 1, p = 0.224) and no differences were found at the basal level (CMR-FT P: CCW n = 6, CW n = 9, VUS: CCW n = 4, CW n = 23, B CCW n = 4, CW n = 8, p = 0.164; Echo-ST P: CCW n = 2, CW n = 7, VUS: CCW n = 1, CW n = 20, B CCW n = 3, CW n = 6, p = 0.094).

Cardiac rotational patterns of the LVNC genetic subgroups and the C population were evaluated using both modalities and are presented in **Fig 3**. In the C group, normal rotational pattern and positive RBR were described using both CMR and echocardiography. In the total LVNC population, normal rotational pattern, positive RBR, negative RBR and reverse rotational pattern were found in 52%, 20%, 22% and 6%, respectively. Similarly to C subjects, LVNC subjects from the B subgroup showed normal rotational patterns and positive RBR with CMR-FT and also with Echo-ST, except for one person with negative RBR. In the VUS and P LVNC subgroups, not only normal rotational pattern and positive RBR were presented, but also approximately one-third to one-fourth of the population had negative RBR with both

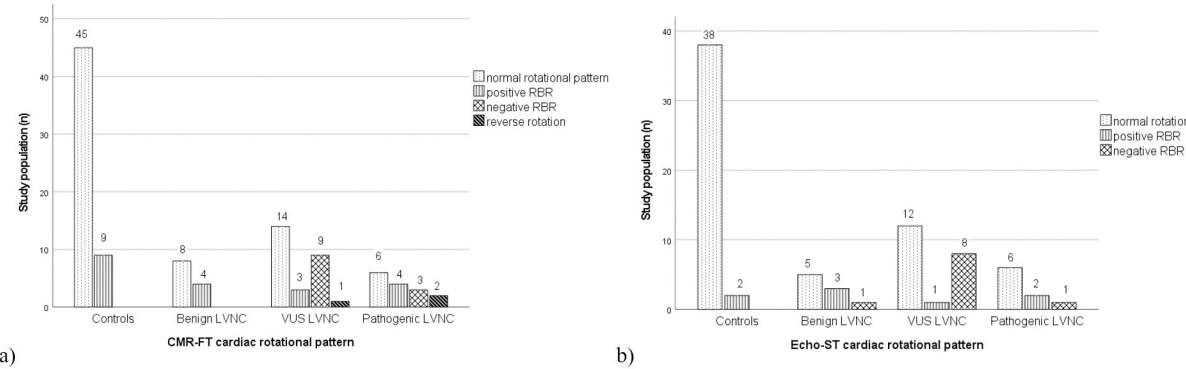

**Fig 3.** Cardiac rotational patterns in LVNC subjects with B, VUS and P genotypes and control individuals measured with CMR-FT (a) and Echo-ST (b) methods. LVNC = left ventricular noncompaction, B = benign, VUS = variant of uncertain significance, P = pathogenic, CMR-FT = cardiac magnetic resonance imaging feature-tracking method, Echo-ST = speckle-tracking echocardiography, RBR = rigid body rotation.

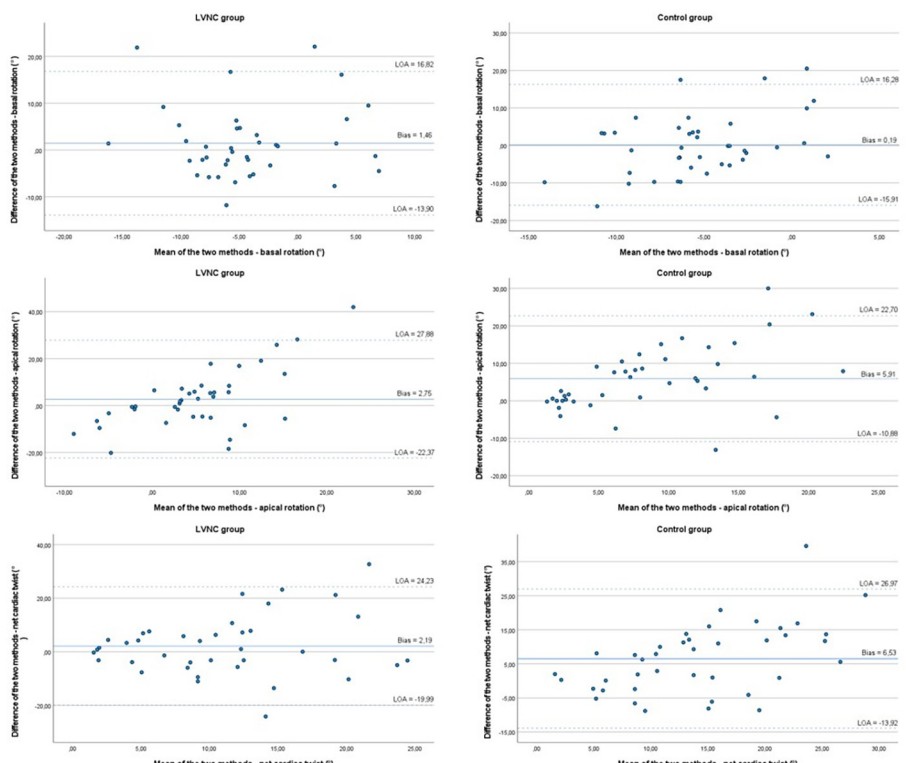

**Fig 4. Comparison of the CMR-FT and Echo-ST rotation degrees in the LVNC and C groups–correlation and Bland-Altman plots with bias and LOA.** CMR-FT = cardiac magnetic resonance imaging feature-tracking method, Echo-ST = speckle-tracking echocardiography, LVNC = left ventricular noncompaction, C = control group, LOA = limit of agreement.

methods. In addition, CMR-FT described a reverse rotational pattern in 1 VUS and 2 P LVNC individuals, where valuable echocardiography images were not available.

Finally, we present the results of intermodality comparisons of CMR-FT and Echo-ST cardiac rotational data in the same individual. Regarding the apical and basal rotational degrees and net cardiac twist, we found no remarkable correlation or reasonable agreement between the two methods in both the total LVNC population and in the C group. Notably, the direction of basal and apical rotation and the cardiac rotational pattern showed moderate to good association between CMR-FT and Echo-ST methods in both the total LVNC (Cohen's kappa: basal 0.65, apical 0.60 and net cardiac twist 0.65, p<0.05) and C groups (Cohen's kappa: basal 0.40, apical 1.0, net cardiac twist 0.40, p<0.05). The results of Bland-Altman and correlation analyses are reported in **Fig 4** and **Table 3**.

## 4. Discussion

In this study, we analyzed cardiac rotation in the light of genetic involvement using CMR-FT and Echo-ST in LVNC morphology subjects with preserved LVEF and healthy controls to evaluate the differences in the degree and pattern of rotation.

When comparing the functional parameters, we found significantly higher volumetric and muscle mass and lower LVEF and apical CLT values in the LVNC group than in controls using both CMR and echocardiography, which is supported by the literature [32–34].

**Table 3. Comparision of the CMR-FT and Echo-ST rotational parameters degrees in the LVNC and C groups–correlations and Bland-Altman analysis.**

| CMR-FT versus Echo-ST | Correlation | | | | Bland-Altman analysis | | | | | |
|---|---|---|---|---|---|---|---|---|---|---|
| | r | p | r | p | Bias | | p | | 95% LOA | |
| | LVNC | | C | | LVNC | C | LVNC | C | LVNC | C |
| **Basal rotation (˚)** | 0.33 | 0.039 | -0.08 | 0.626 | 1.46 | 0.19 | 0.243 | 0.884 | -13.90; 16.82 | -15.91; 16.28 |
| **Apical rotation (˚)** | 0.11 | 0.496 | 0.34 | 0.031 | 2.75 | 5.91 | 0.179 | <0.001* | -22.37; 27.88 | -10.88; 22.70 |
| **Net cardiac twist (˚)** | 0.15 | 0.366 | 0.36 | 0.022 | 2.19 | 6.53 | 0.239 | <0.001* | -19.99; 24.23 | -13.92; 26.97 |

CMR-FT = cardiac magnetic resonance imaging feature-tracking method, Echo-ST = speckle-tracking echocardiography, LVNC = left ventricular noncompaction, C = control group, Bias = the mean value of the difference between the CMR-FT and Echo-ST methods, LOA = limit of agreement, the mean value of the difference between the two methods ± 2SD, * = p<0.05

## 4.1 Rotational parameters in the LVNC and C groups

Our results showed no differences in basal rotation between the total LVNC population and C group. Similar results were described in the literature [10, 12, 31, 35, 36]; however, some studies mention that patients with heart failure had a reduced basal rotational degree [8, 11, 37].

In our study, we described reduced apical rotational degrees and net cardiac twist in the LVNC group with preserved LVEF than in C individuals; furthermore, negative CW apical rotation was presented only in LVNC subjects. These findings were previously described in heart failure patients, individuals with hypertrabeculation without meeting the criteria of LVNC, relatives of LVNC patients [8, 9, 11, 12, 15, 31, 36]; and interestingly reduced apical rotational degree was also described in elite endurance athletes [38]. Guigui et al reported no differences in net cardiac twist between hypertrabeculated and control subjects; however, it is worth noting that more than half of their LVNC study population was of African-American ethnicity, which may have influenced their findings [14, 39].

A study in animal models described that apical rotation and cardiac twist dose-dependently increased after dobutamine and decreased after esmolol infusion; while basal rotation remained unchanged [40]. Moreover, they found that apical rotation was more closely related to contractility (dP/dt) than LVEF [40], and other studies underline the influence of twist on cardiac function in patients with heart failure [8, 15, 37]. These findings suggest that the rotation of the subendocardial fibers may play a special role in maintaining the mechanical function of the heart, and thus abnormal rotation as a subclinical sign may be a possible predictor of contractility impairment resulting in structural-functional remodeling of the heart.

The relationship between hypertrabeculation and rotational changes can be emphasized with the negative correlation of apical and net rotation with LVTPMi and volumetric parameters in our study. The identification of apical rotation as an independent predictor of cardiovascular complications [41] and apical strain deterioration in hypertrabeculated patients with heart failure [10, 34] also underline the necessity of apical movement evaluation in clinical practice.

## 4.2 Analyzing LVNC genetic subgroups

Comparing the three genetic LVNC subgroups, we found no significant differences in volumetric and LVEF parameters measured both with CMR and echocardiography. These connections were not investigated before in LVNC subjects with preserved LVEF, while other studies mention a connection between impaired LVEF and genotype [42, 43]. Interestingly, the mid and apical segment CLT were reduced in individuals with genetic involvement, and a recent study reported the connection between CLT<5 mm in these segments and LVNC with

reduced LVEF. The differences between P, VUS and B subgroups in clinical manifestation are in line with our previous research data [44].

Regarding cardiac rotation, basal and apical rotational degrees and net cardiac twist were comparable among the three LVNC subgroups. However, significant differences were found in the direction of apical rotation: negative CW apical rotation was reported only in subjects with genetic involvement. To the best of our knowledge, no previous similar study comparing cardiac rotation in LVNC morphology subjects with different genotype was conducted.

### 4.3 Cardiac rotational patterns

In terms of cardiac rotational patterns in the C group, besides the normal variant, positive RBR was also presented in a small proportion. The occurrence of positive RBR in otherwise healthy individuals had been previously reported in the literature without understanding its significance [10, 14, 45].

In terms of the total LVNC population, approximately half of them had a normal rotational pattern, one-fifth positive RBR and the remaining 30% negative RBR and reverse rotational pattern. This is in line with studies conducted on LVNC populations including both preserved and reduced LVEF patients [10, 11, 14, 15, 31, 36]. Interestingly, van Dalen et al described mostly negative RBR in their total LVNC population with heart failure, raising the possibility that negative RBR might be more likely associated with poor LV function [12]. Negative RBR has also been suggested and highlighted in the literature as a hallmark of LV hypertrabecula-tion, as reversed apical rotation was linked with cardiovascular complications, a more severe stage of heart failure and myocardial fibrosis [11, 16, 46]. Interestingly, longer QRS interval, higher occurrence of left bundle branch block and lower LVEF were also observed in LVNC patients with negative RBR than in those with positive RBR [12, 13].

Regarding the LVNC genetic subgroups, we found a normal rotational pattern in two-thirds and positive RBR in the remaining one-third of subjects with B genotype. In the VUS and P subgroups, RBR was present in nearly half of the population with a predominantly nega-tive pattern, and approximately 10% had reverse rotation. No previous studies in the literature analyzed cardiac rotational patterns in LVNC subjects with different genotypes. However, van Dalen et al reported an abnormal rotational pattern in all LVNC subjects with a positive family history; and RBR was described in approximately one-third of first-degree relatives of LVNC patients and also in hypertrabeculation with positive family history [13, 15, 36].

### 4.4 The intermodality agreement between CMR-FT and Echo-ST regarding rotational parameters

In terms of basal and apical rotational degrees and net cardiac twist measured with CMR-FT and Echo-ST methods, no remarkable correlation or reasonable agreement was found and CMR-FT calculated higher values. These results are supported by the literature on the overesti-mation of CMR-FT and underestimation of Echo-ST methods compared both to tagging imaging [5, 29]. However, a study comparing the results of cardiac rotation measurements per-formed with CMR-FT and Echo-ST in hypertrabeculated subjects was not conducted previ-ously. Similar to our results, there was no good agreement between the two methods for quantitative strain analysis either in the literature or in our pilot LVNC study [47, 48]. On the contrary, these modalities correlated well in terms of volume measurement [49].

In our study, the difference in rotational degrees between the two modalities could be explained by the technical background, as CMR-FT follows the endocardial border in three directions and Echo-ST tracks the intramyocardial features in two dimensions [6].

When evaluating the direction and pattern of rotation, we found moderate to good agreement between CMR-FT and Echo-ST techniques, indicating the wider usability of these parameters regardless of the measuring modality. To the best of our knowledge, there are no similar comparisons available in the literature.

Although in clinical practice, cardiac rotation could be evaluated by both CMR-FT and Echo-ST techniques, caution is required, as these modalities may not be interchangeable. Therefore, follow-up examinations and serial measurements are only comparable within the same modality.

In summary, the decrease in apical rotation and the consequent negative RBR may be a consequence of mechanical changes caused by hypertrabecularisation, especially in LVNC subjects with genetic involvement. Therefore, cardiac rotation might be an early indicator of deteriorating cardiac function and might be a warning sign in clinical follow-up conducted by either CMR or cardiac ultrasound modalities; however, further investigations are needed.

## 5. Conclusion

In our study, we analyzed the cardiac rotation in hypertrabeculated phenotype subjects with preserved LVEF and C individuals and evaluated whether genetic involvement might influence the pattern of rotation.

The degree of apical rotation and net cardiac twist were significantly reduced in the total LVNC group compared to controls; showed a significant correlation with LV volumes and LVTPMi; and were comparable between the three genetic subgroups.

The direction of apical rotation differed significantly among different genotype LVNC subjects: a negative apical rotation was detected only in persons with P and VUS genotypes. When examining the pattern of rotation, we found normal rotation and positive RBR in B genotype LVNC and C persons, whereas negative RBR was detected in a significant proportion of P and VUS genotype LVNC phenotype subjects.

In contrast to the degree, the direction and pattern of cardiac rotation were comparable between the CMR-FT and Echo-ST methods.

To conclude, our above-mentioned novelties suggest that apical rotation could be affected in hypertrabeculation, especially in subjects with genetic involvement. This might be a sensitive marker for subclinical contractility impairment; thus further clinical investigations may be required.

## 6. Limitations

This is a single-center retrospective cardiac imaging-focused study, thus clinical data and follow-up of participants were limited. As primary LVNC morphology with genetic involvement is a low-prevalence phenomenon, the small sample size of the genetic subgroups may have affected the statistical results, and the refreshing genetic database could modify the VUS classification. When analyzing with Echo-ST, the smaller sample size may explain the tendentious differences in some results without reaching the significance threshold.

Finally, we have to mention the limitations of the measuring methods. The TB method is influenced by the actual path of trabeculae and papillary muscles. Although CMR-FT and Echo-ST are both validated methods for clinical research, it is important to ensure protocol adherence and use by an experienced observer for good reproducibility.

## Supporting information

**S1 Table. The identified CMP-related P, LP and VUS mutations in our LVNC study population.** CMP = cardiomyopathy, LVNC = left ventricular noncompaction, P = pathogenic,

LP = likely pathogenic, VUS = variant of uncertain significance, ID = identifier,
HGVSc = human genome variation society coding DNA sequence, HGVSp = human genome
variation society protein sequence.
(DOCX)

**S2 Table. Interobserver agreement (ICC) for the measured functional and rotational
parameters with both CMR and echocardiography.** ICC = intraclass correlation: average
measures (95% confidential interval lower and upper band), CMR = cardiac magnetic reso-
nance imaging, LVEDVi = left ventricular end-diastolic volume index, LVESVi = left ventricu-
lar end-systolic volume index, LVSVi = left ventricular stroke volume index, LVEF = left
ventricular ejection fraction, LVTMi = left ventricular total mass index, LVTPMi = left ventric-
ular trabeculated and papillary muscle mass index, NA = not applicable.
(DOCX)

**S1 Dataset.**
(XLSX)

## Acknowledgments

We would like to thank the technicians who helped during the cardiac magnetic resonance
imaging examinations.

## Author Contributions

**Conceptualization:** Andrea Szűcs.

**Data curation:** Kinga Grebur, Balázs Mester, Márton Horváth, Kristóf Farkas-Sütő, Zsófia
Gregor, Anna Réka Kiss, Alexandra Fábián, Bálint Károly Lakatos, Katalin Csonka.

**Formal analysis:** Kinga Grebur.

**Investigation:** Kinga Grebur, Andrea Szűcs.

**Methodology:** Kinga Grebur, Attila Tóth, Attila Kovács, Bálint András Fekete, Csaba Bödör,
Andrea Szűcs.

**Project administration:** Andrea Szűcs.

**Resources:** Kinga Grebur.

**Supervision:** Béla Merkely, Hajnalka Vágó, Andrea Szűcs.

**Visualization:** Kinga Grebur.

**Writing – original draft:** Kinga Grebur.

**Writing – review & editing:** Zsófia Gregor, Anna Réka Kiss, Attila Tóth, Attila Kovács, Bálint
András Fekete, Csaba Bödör, Hajnalka Vágó, Andrea Szűcs.

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
