## [Decision Letter · Decision Letter 0]

2 May 2024

PONE-D-24-01299The effect of excessive trabeculation on cardiac rotation – a multimodal imaging studyPLOS ONE

Dear Dr. Szűcs,

Thank you for submitting your manuscript to PLOS ONE. After careful consideration, we feel that it has merit but does not fully meet PLOS ONE’s publication criteria as it currently stands. Therefore, we invite you to submit a revised version of the manuscript that addresses the points raised during the review process.

We look forward to receiving your revised manuscript.

Kind regards,

Yashendra Sethi

Academic Editor

PLOS ONE

Journal Requirements:

Additional Editor Comments:

Dear authors,

Thank you for the submission. The reviewers have now evaluated your manuscript and have raised some concerns which need to addressed before we can proceed further.

Reviewers' comments:

Reviewer's Responses to Questions

**Comments to the Author**

1. Is the manuscript technically sound, and do the data support the conclusions?

Reviewer #1: Yes

Reviewer #2: Partly

2. Has the statistical analysis been performed appropriately and rigorously? 

Reviewer #1: Yes

Reviewer #2: Yes

3. Have the authors made all data underlying the findings in their manuscript fully available?

Reviewer #1: Yes

Reviewer #2: Yes

4. Is the manuscript presented in an intelligible fashion and written in standard English?

Reviewer #1: Yes

Reviewer #2: Yes

5. Review Comments to the Author

Reviewer #1: Dear Authors

Congratulations to your nice paper written in Plos One. The paper has an important scientific message. The manuscript is interesting and the idea is new. The manuscript can be accepted after minor revision. Please provide the p values in the abstract.

Reviewer #2: In this study, the authors compared a group of individuals with LV hypertrabeculation in the absence of ventricular dysfunction who underwent both CMR and genetic testing to a group of controls in terms of LV patterns and degree of rotation. They found significant differnces in the degree of apical, but not basal, rotation between LVNC and C. Pattenrs of rotation also differed between groups: while in all controls the apex rotate in a clockwise fashion (i.e. normal cardiac rotation or positive RBR), in 15/54 subjects with NLVC the apex rotated in a counterclockwise fashion (negative RBR or reverse cardiac rotation). OF note, when dividing NLVC subjects according to genetic testing results, counter clockwise apical rotation occurred only in those with VUS or P variants, but not in those with completely negative testing.

The authors concluded that abnormal apical rotation may be a sensitive marker for cardiac contractility impairment.

General comment: the authors addressed a very controversial topic, which is the clinical significance of isolated hypertrabeculation. True LVNC is a rare condition, a congenital defect caused by anomalous development of the cardiac muscle. Much more common is the benign hypertrabeculation, in which there is an increased C/NC ratio (Petersen's criteria) but with preserved thickness of the compact layer and preserved ventricular contractility. This condition is a variant of normal, which can also develop because of increased cardiac load such as pregnancy and training. Also the results of genetic testing should be interpreted with caution, particularly in a disease such as LVNC in which the genetic aspect remains elusive.

Specific comments:

1) In the study, it is unclear whether patients were affected by a disease or simply by a phenotypic variant. What was the cause for the MRI? Was there any other clinical manifestation (e.g. ventricular arrhythmias, positive family history...) suggesting a true cardiac disease? And most importantly, was the thickness of the compacted layer reduced or preserved (a cut-off of 5 mm has recently been proposed for MRI)? I think better characterization of the study sample is needed. I really think comparison between subjects with NLVC and decrease thickness of the compact layer versus those with LVNC but normal thickness of the compact layer would make much sense

2) Genetic testing: the authors tested a panel of 174 genes but did not report the results in details. We only know that in 15 a pathogenetic mutation was found and in 27 a VUS, but in which genes? Of course, when using such a large panel, clinical plausibility is of paramount importance when interpreting the results. For example, a mutation in the SCN5A gene, even if pathogenetic, is very unlikely to be linked to NLVC. On the other hand, mutations in genes linked to LVNC such as the MYH7 or MYBPC3 gene have much higher relevance.

3) clinical interpretation: the findings of the study suggest that in a subset of patients with NLVC the apex rotates differently than controls, but that does not necessary imply a disease. As stated in the discussion, the same pattern has also been described in very healthy and fit individuals (elite athletes). It is possible that hypertrabeculation of the LV apex causes the apex to rotate differently but this does not will translate into an overt ventricular dysfunction later in the life course. As a consequence, I believe your conclusions about the clinical implications of this finding should be tempered down. In my opinion, this observation remains... an observation, but its meaning remains to be established.

6. PLOS authors have the option to publish the peer review history of their article (what does this mean?). If published, this will include your full peer review and any attached files.

Reviewer #1: No

Reviewer #2: No

---

## [Author Response · Author response to Decision Letter 0]

19 Jun 2024

Reviewer comments and answers

Reviewer #1: Dear Authors

Congratulations to your nice paper written in Plos One. The paper has an important scientific message. The manuscript is interesting and the idea is new. The manuscript can be accepted after minor revision. Please provide the p values in the abstract.

We would like to thank the reviewer for the appreciation! We provided the missing p values in the abstract in line 40,43,47. 

Reviewer #2: In this study, the authors compared a group of individuals with LV hypertrabeculation in the absence of ventricular dysfunction who underwent both CMR and genetic testing to a group of controls in terms of LV patterns and degree of rotation. They found significant differnces in the degree of apical, but not basal, rotation between LVNC and C. Pattenrs of rotation also differed between groups: while in all controls the apex rotate in a clockwise fashion (i.e. normal cardiac rotation or positive RBR), in 15/54 subjects with NLVC the apex rotated in a counterclockwise fashion (negative RBR or reverse cardiac rotation). OF note, when dividing NLVC subjects according to genetic testing results, counter clockwise apical rotation occurred only in those with VUS or P variants, but not in those with completely negative testing.

The authors concluded that abnormal apical rotation may be a sensitive marker for cardiac contractility impairment.

General comment: the authors addressed a very controversial topic, which is the clinical significance of isolated hypertrabeculation. True LVNC is a rare condition, a congenital defect caused by anomalous development of the cardiac muscle. Much more common is the benign hypertrabeculation, in which there is an increased C/NC ratio (Petersen's criteria) but with preserved thickness of the compact layer and preserved ventricular contractility. This condition is a variant of normal, which can also develop because of increased cardiac load such as pregnancy and training. Also the results of genetic testing should be interpreted with caution, particularly in a disease such as LVNC in which the genetic aspect remains elusive.

Specific comments:

1) In the study, it is unclear whether patients were affected by a disease or simply by a phenotypic variant. What was the cause for the MRI? Was there any other clinical manifestation (e.g. ventricular arrhythmias, positive family history) suggesting a true cardiac disease? And most importantly, was the thickness of the compacted layer reduced or preserved (a cut-off of 5 mm has recently been proposed for MRI)? I think better characterization of the study sample is needed. I really think comparison between subjects with NLVC and decrease thickness of the compact layer versus those with LVNC but normal thickness of the compact layer would make much sense

We would like to thank the reviewer for the suggestions; they helped improve our manuscript. 

Left ventricular noncompaction (LVNC), hypertrabeculation or excessive trabeculation has become the subject of renewed interest since the Petersen et al statement and the new ESC cardiomyopathy guideline published in 2023 [1, 2]. These modifications raise novel scientific questions and inspire researchers to investigate the noncompacted phenotype in more detail, especially in cases with preserved cardiac function. 

Regarding the study enrollment, in our highly progressive clinic, we have been monitoring the patient population since 2008, highlighting and following hypertrabeculated morphology subjects. The indication of CMR was either the suspicion of LVNC raised by echocardiography, family history and cardiac symptoms, or the CMR finding was incidental. In this present study, we included LVNC individuals from this database with preserved left ventricular ejection fraction (LVEF), clinical manifestation and available cardiogenetic data (see details in our Methods section). We excluded individuals with secondary or transient hypertrabeculation and a sport activity >6 hours/week [1, 3] 

All of the enrolled LVNC subjects had clinical manifestations, namely cardiac symptoms and/or positive family history. The reviewer was right about this missing information; we included it in our manuscript in Table1a, Results section lines 324-326 and Discussion section 428-429. 

 Pathogenic subgroup (n=15) VUS subgroup (n=27) Benign subgroup (n=12) p

Positive family history (n) 10 13 1 0.008

Atypical chest pain (n) 6 6 8 0.03

Palpitation (n) 8 14 6 1.00

Syncope (n) 2 2 1 0.834

Arrhythmia (n) 8 18 7 0.632

As the reviewer pointed out, a recent study highlighted the importance of measuring the compact layer thickness (CLT) in LVNC. This novel finding suggests the prognostic role of CLT in differentiating hypertrabeculation from noncompaction with decreased cardiac function [4].

Comparing the CLT in the LVNC and control (C) groups, only the apical segment differed significantly (apical CLT LVNC 4.77±0.75 mm, C 5.44±0.23 mm, p<0.001). Differences were found in the mid and apical CLT between the pathogenic (P), variant of uncertain significance (VUS) and benign (B) subgroups (mid CLT P 5.29±0.87 mm, VUS 5.91±0.53 mm, B 5.93±0.51 mm, p=0.009; apical CLT P 3.9±0.65 mm, VUS 5.01±0.52 mm, B 5.32±0.28 mm, p<0.001), while it was comparable in the basal segment. The CLT parameters were included in Table 1b in the Results section of the manuscript. 

Regarding the relationship between the CLT and cardiac rotational parameters, no significant correlation was found in the LVNC and C groups. See details in Table 2 in the Results section. However, it is interesting to note that the mid and apical CLT showed a significant moderate to strong correlation with genotype: mid CLT r=-0.351, p=0.009; apical CLT r=-0.691, p<0.001.

With the abovementioned details we completed the Methods, the Results and the Discussion sections of the manuscript in lines 172-174, 262, 322-324, 391 and 426-428. 

According to the reviewer’s suggestion, based on the CLT we divided the LVNC group into two subgroups: 28 person had a minimum of one segment with reduced CLT (CR; CLT<5mm) and 26 subjects had normal CLT(CN; CLT>5mm) in basal, mid and apical segments. Comparing the rotational parameters between CR and CN subgroups, we found no significant differences regarding the rotational degrees and net cardiac twist. See details below.

 CR subgroup

(CLT<5 mm) CN subgroup

(CLT>5 mm) p

Basal rotational degree (°) -1.7±7.2 -5.4±6.5 p=0.053

Apical rotational degree (°) 5.2±11.2 8.2±10.8 p=0.316

Net cardiac rotation (°) 10.2±9.1 14.7±9.9 p=0.085

Similarly, the direction of basal and apical rotation was comparable between the CR and CN subgroups (basal rotation CR 18 negative, 10 positive, CN 22 negative, 4 positive, p=0.124; apical rotation CR 11 negative, 17 positive, CN 4 negative, 22 positive; p=0.07). Interestingly, negative RBR was more frequent in the CR subgroup and reverse rotation occurred only in this subgroup, although this result was not significant statistically: CR subgroup 10 normal rotation, 7 positive RBR, 8 negative RBR, 3 reverse rotation; CN subgroup 18 normal rotation, 4 positive RBR, 4 negative RBR, 0 reverse rotation, p=0.061.

Our study was designed to analyze the rotational parameters of LVNC with preserved cardiac function in relation to genetic data, as our clinical experience and previous research pointed out that genotype, phenotype and clinical manifestation can be related [5]. Thus, we did not modify the original genotype-based focus of LVNC individuals in our manuscript; however, we included the abovementioned major results of CLT measurement, except the CLT-based classification. Meanwhile, if the reviewer considers it, we might include this topic as well. 

We would like to thank the reviewer for the novel aspect and hope it could be a topic of further investigations. 

2) Genetic testing: the authors tested a panel of 174 genes but did not report the results in details. We only know that in 15 a pathogenetic mutation was found and in 27 a VUS, but in which genes? Of course, when using such a large panel, clinical plausibility is of paramount importance when interpreting the results. For example, a mutation in the SCN5A gene, even if pathogenetic, is very unlikely to be linked to NLVC. On the other hand, mutations in genes linked to LVNC such as the MYH7 or MYBPC3 gene have much higher relevance.

The reviewer is absolutely right about the importance of reporting the complex genetic data of our LVNC study group. We reported in detail the pathogenic and VUS mutations in Supplementary Table 1. Mutations were considered relevant if they were previously associated with LVNC or with other cardiomyopathies due to genetic overlap [6].

3) clinical interpretation: the findings of the study suggest that in a subset of patients with NLVC the apex rotates differently than controls, but that does not necessary imply a disease. As stated in the discussion, the same pattern has also been described in very healthy and fit individuals (elite athletes). It is possible that hypertrabeculation of the LV apex causes the apex to rotate differently but this does not will translate into an overt ventricular dysfunction later in the life course. As a consequence, I believe your conclusions about the clinical implications of this finding should be tempered down. In my opinion, this observation remains... an observation, but its meaning remains to be established.

Regarding the third comment, we would like to apologize for any misunderstanding in the discussion section. Some publications analyzing cardiac rotation in elite athletes describe decreased apical rotational degree and net cardiac twist compared to controls, while others report no differences between the populations [7]. In contrast to these athletes, our genetically involved LVNC study population and some types of cardiomyopathies were characterized by negative apical rotation [8-10]. At the same time, it is possible that the hypertrabeculed morphology itself contributes to the change in cardiac rotation; however, it is interesting to note that turning apical rotation from positive to negative was only present in genetically affected individuals, and not in the benign subgroup and controls. Thus we corrected the Discussion in lines 402-403. 

Overall, we agree that cardiac rotation should be interpreted together with other parameters such as morphology, functional parameters, clinical manifestation, family history and genetic data. As the reviewer states and as we have mentioned in the Conclusion section further investigations are required regarding its clinical implication in risk stratification, as the connection between negative apical rotation and positive genotype strengthened its subclinical role.

According to these aspects, we modified the Discussion and Conclusion sections in lines 486-489 and 506. 

We hope our improvements meet the reviewer requirements and have increased the value of the manuscript.

References

1. Petersen, S.E., et al., Excessive Trabeculation of the Left Ventricle: JACC: Cardiovascular Imaging Expert Panel Paper. JACC Cardiovasc Imaging, 2023. 16(3): p. 408-425.

2. Arbelo, E., et al., 2023 ESC Guidelines for the management of cardiomyopathies. Eur Heart J, 2023.

3. Pelliccia, A., et al., 2020 ESC Guidelines on Sports Cardiology and Exercise in Patients with Cardiovascular Disease. Rev Esp Cardiol (Engl Ed), 2021. 74(6): p. 545.

4. De Lazzari, M., et al., Thinning of compact layer and systolic dysfunction in isolated left ventricular non-compaction: A cardiac magnetic resonance study. Int J Cardiol, 2024. 397: p. 131614.

5. Grebur, K., et al., Genetic, clinical and imaging implications of a noncompaction phenotype population with preserved ejection fraction. Front Cardiovasc Med, 2024. 11: p. 1337378.

6. Mazzarotto, F., et al., Systematic large-scale assessment of the genetic architecture of left ventricular noncompaction reveals diverse etiologies. Genet Med, 2021. 23(5): p. 856-864.

7. Beaumont, A., et al., Left Ventricular Speckle Tracking-Derived Cardiac Strain and Cardiac Twist Mechanics in Athletes: A Systematic Review and Meta-Analysis of Controlled Studies. Sports Med, 2017. 47(6): p. 1145-1170.

8. Popescu, B.A., et al., Left ventricular remodelling and torsional dynamics in dilated cardiomyopathy: reversed apical rotation as a marker of disease severity. Eur J Heart Fail, 2009. 11(10): p. 945-51.

9. Peters, F., et al., Left ventricular twist in left ventricular noncompaction. Eur Heart J Cardiovasc Imaging, 2014. 15(1): p. 48-55.

10. van Dalen, B.M., et al., Diagnostic value of rigid body rotation in noncompaction cardiomyopathy. J Am Soc Echocardiogr, 2011. 24(5): p. 548-55.

---

## [Decision Letter · Decision Letter 1]

17 Jul 2024

The effect of excessive trabeculation on cardiac rotation – a multimodal imaging study

PONE-D-24-01299R1

Dear Dr. Szűcs,

We’re pleased to inform you that your manuscript has been judged scientifically suitable for publication and will be formally accepted for publication once it meets all outstanding technical requirements.

Kind regards,

Yashendra Sethi

Academic Editor

PLOS ONE

Additional Editor Comments (optional):

Congratulations!

Reviewers' comments:

Reviewer's Responses to Questions

**Comments to the Author**

1. If the authors have adequately addressed your comments raised in a previous round of review and you feel that this manuscript is now acceptable for publication, you may indicate that here to bypass the “Comments to the Author” section, enter your conflict of interest statement in the “Confidential to Editor” section, and submit your "Accept" recommendation.

Reviewer #1: All comments have been addressed

Reviewer #2: All comments have been addressed

2. Is the manuscript technically sound, and do the data support the conclusions?

Reviewer #1: Yes

Reviewer #2: Yes

3. Has the statistical analysis been performed appropriately and rigorously? 

Reviewer #1: Yes

Reviewer #2: Yes

4. Have the authors made all data underlying the findings in their manuscript fully available?

Reviewer #1: Yes

Reviewer #2: Yes

5. Is the manuscript presented in an intelligible fashion and written in standard English?

Reviewer #1: Yes

Reviewer #2: Yes

6. Review Comments to the Author

Reviewer #1: Dear Authors

Congratulations to your nice paper written in Plos One. The paper has an important scientific message. The manuscript is interesting and the idea is new. The manuscript can be accepted for publication.

Regards

Reviewer #2: (No Response)

7. PLOS authors have the option to publish the peer review history of their article (what does this mean?). If published, this will include your full peer review and any attached files.

Reviewer #1: No

Reviewer #2: **Yes: **Alessandro Zorzi

---

## [Editor Report · Acceptance letter]

25 Jul 2024

PONE-D-24-01299R1 

PLOS ONE

Dear Dr. Szűcs, 

I'm pleased to inform you that your manuscript has been deemed suitable for publication in PLOS ONE. Congratulations! Your manuscript is now being handed over to our production team.

Kind regards, 

on behalf of

Dr. Yashendra Sethi 

Academic Editor

PLOS ONE